# Exploring the Risk Factors for Poor Survival in Lupus Pericarditis Patients: A Retrospective Cohort Study

**DOI:** 10.3390/jcm11185473

**Published:** 2022-09-17

**Authors:** Yen-Fu Chen, Meng-Ru Hsieh, Che-Tzu Chang, Ping-Han Tsai, Yao-Fan Fang

**Affiliations:** 1Division of Rheumatology, Allergy and Immunology, Department of Internal Medicine, Chang Gung Memorial Hospital, Taoyuan City 333, Taiwan; 2Division of Rheumatology, Allergy and Immunology, New Taipei Municipal TuCheng Hospital, New Taipei City 236, Taiwan

**Keywords:** systemic lupus erythematosus, pericarditis, mortality

## Abstract

Patients with systemic lupus erythematosus (SLE) have a higher risk of pericarditis, which could be fatal. The goal of this study was to identify the prognostic factors for mortality in patients with lupus pericarditis. Patients with lupus pericarditis treated at Chang Gung Memorial Hospital were included in this observational cohort study. This study conducted univariate and multivariate COX regression, as well as Kaplan–Meier survival curve analysis, to investigate mortality risk in SLE patients. The average age at admission was 40.78 ± 15.92 years. A total of 113 (16.4%) of the 689 patients had lupus pericarditis. Patients with lupus pericarditis exhibited older age, shorter follow-up, higher disease activities, and higher incidence rates of comorbidities than patients without pericarditis. Cox regression adjusted analysis indicated that lupus pericarditis (hazard ratio = 1.963, 95% CI = 1.315, 2.963, *p* = 0.001), old age at admission (HR = 1.053, 95% CI = 1.040, 1.065, *p* < 0.001), high SLEDAI score (HR = 1.079, 95% CI = 1.043, 1.116, *p* < 0.001), and end-stage kidney disease (ESKD) (HR = 2.533, 95% CI = 1.620, 3.961, *p* < 0.001) were all linked to increased mortality. Moreover, the Kaplan–Meier survival curve analysis revealed that patients with pericarditis compared to those without pericarditis had a higher mortality rate (log-rank test, *p* < 0.001). A high proportion of SLE patients have manifestations of lupus pericarditis. Moreover, patients with lupus pericarditis have a greater risk of mortality even if they have no pericardial tamponade. Therefore, these patients need prompt diagnosis and treatment.

## 1. Introduction

Systemic lupus erythematosus (SLE) is a chronic systemic inflammatory autoimmune disorder. The clinical features of SLE patients can vary from mild joint and skin involvement to severe life-threatening internal organ diseases. Inflammation of the serous membrane (e.g., the pleura and pericardium) is one of the 11 American College of Rheumatology (ACR) Criteria for SLE classification, and it may result in pain or fluid accumulation [1]. SLE may affect all parts of the heart, including the pericardium, conduction system, myocardium, valves, and coronary arteries [2]. Pericarditis is also the most common cardiac disorder in SLE [2,3,4,5,6]. Pericardial disorders are common in patients with connective tissue diseases and may cause a wide range of symptoms, such as from acute pericarditis to cardiac tamponade [7,8]. Pericardial involvement has been found to be common in SLE, rheumatoid arthritis, and scleroderma, but not in inflammatory myositis [3]. Additionally, pericardial involvement may be the initial manifestation of SLE in a small number of patients [9,10].

The development of lupus pericarditis along with cardiac tamponade is associated with high mortality in SLE patients [6,9]. However, limited studies have been conducted on the prevalence, prognostic factors, and survival outcomes of lupus pericarditis patients. In addition, previous studies have mostly found that patients with cardiac tamponade have a higher mortality rate [6,9]. Nevertheless, our study found that lupus pericarditis could be a severe inflammatory alteration that contributed to a higher risk of mortality. In this study, the data were collected through the examination of the medical records of our SLE patients admitted to the rheumatology ward and the assessment of clinical aspects, as well as defined prognostic factors in lupus pericarditis patients.

## 2. Materials and Methods

### 2.1. Study Population

Patients with SLE who were admitted to and treated at Chang Gung Memorial Hospital between January 2005 and December 2012 were enrolled into the study and were followed up until 31 March 2019. The diagnosis of SLE was based on the 1997 American College of Rheumatology Criteria (ACR1997) [1]. The Institutional Review Board of Chang Gung Medical Foundation (103-2394C) exempted the need for informed consent from participants in this study because the original identification numbers of each patient in the database were encrypted.

### 2.2. Data Collection

The medical records of 689 patients with SLE who were admitted to Chang Gung Memorial Hospital in Taiwan were reviewed retrospectively. These medical records were reviewed from the time of diagnosis until death, loss of follow-up, or until the end of the study (31 March 2019). The collected data included sex, age, clinical presentations, laboratorial test results, comorbidities, and causes of mortality. The lupus pericarditis patients’ medical records were kept updated since their admission. Patients with infections, malignancies, and pericarditis caused by heart failure were excluded from this study. Lupus pericarditis was diagnosed when any of the following clinical features were present in patients: typical sharp precordial pains; pericardial rubs; electrocardiographic abnormalities of pericarditis or pericardial effusion; and diagnostic signs of enlargement of the cardiac silhouette on a chest radiograph together with the echocardiographic diagnosis of pericardial effusion, as well as exclusion of other causes of pericarditis, such as tuberculosis or malignancy [9].

### 2.3. Statistical Analysis

Categorical data are expressed as percentages, whereas numerical data are presented as means and standard deviations (SDs). In group comparisons, the chi-square test was used to compare categorical data, whereas the two-tailed Student’s *t*-test was used to compare numerical data. The study results were considered statistically significant at *p* < 0.05. Additionally, multivariate analysis was performed using Cox regression to identify independent risk factors for mortality. The hazard ratio (HR) was presented along with a 95% confidence interval (CI). Kaplan–Meier curves were used to describe the survival characteristics of SLE patients. All statistical analyses were performed using SPSS 25.0 software (IBM Corp., Armonk, NY, USA).

## 3. Results

### 3.1. Patient Baseline Characteristics

Table 1 shows the demographic characteristics of the 689 hospitalized SLE patients (611 women and 78 men), as well as clinical manifestations, laboratory findings, comorbidities, drug treatment, and comparison between patients with and without lupus pericarditis. The average age at admission was 40.78 ± 15.92 years; the proportion of female patients was 88.70%; and the mean follow-up duration since admission to the hospital was 7.31 ± 5.11 years. The average systemic lupus erythematosus disease activity index (SLEDAI) score was 12.92 ± 6.34. The proportions of patients with diabetes, hypertension, a history of cardiovascular events, and end-stage kidney disease were 7.3%, 31.9%, 8.0%, and 9.6%, respectively. More than 98% of the patients used steroids and hydroxychloroquine as treatment. Moreover, 113 (16.4%) of the 689 SLE patients had lupus pericarditis. Patients with lupus pericarditis exhibited older age and had a shorter follow-up duration. According to disease activity and the laboratory test results, patients with lupus pericarditis had higher SLEDAI scores, lower hemoglobin counts, and lower levels of complement 3 (C3).

Furthermore, patients with pericarditis mostly had cardiovascular events and end-stage kidney disease—the most common comorbidities in pericarditis. The proportions of medications used in both groups were almost the same.

### 3.2. Mortality and Survival Analyses

During the observation period, 129 (18.7%) of the 689 SLE patients died. In the univariate analysis, old age at admission, high SLEDAI scores, low hemoglobin counts, low C3 levels, and the presence of pericarditis, cerebrovascular events, and end-stage renal disease were the risk factors that had a correlation with mortality (Table 2). Patients with lupus pericarditis had a higher mortality rate (31.9%; 36/113) than those without (16.1%; 93/576). Four of the 113 patients were represented with cardiac tamponade as the presentation of systemic lupus erythematosus during the observation period. The causes of mortality were infection in three cases and only one case died from cardiac tamponade-induced sudden death. Moreover, according to the multivariate Cox regression analysis, lupus pericarditis (HR = 1.963, 95% CI = 1.315, 2.963, *p* < 0.001), old age at admission (HR = 1.053, 95% CI = 1.040, 1.065, *p* < 0.001), high SLEDAI scores (HR = 1.079, 95% CI = 1.043, 1.116, *p* < 0.001), and end-stage kidney disease (HR = 2.533, 95% CI = 1.620, 3.961, *p* < 0.001) were associated with higher mortality rates (Table 2).

Figure 1 depicts the survival curve of patients with or without lupus pericarditis. Survival outcomes were significantly lower in patients with pericarditis than those in patients without pericarditis (log-rank test, *p* < 0.001). Patient survival was inversely correlated with the presence of lupus pericarditis. The non-pericarditis group of patients had cumulative survival rates of 93.1%, 91.1%, 86.1%, and 80.7% after 1, 2, 5, and 10 years, respectively. The overall survival rates of the lupus pericarditis patients were 92.8%, 77.8%, 68.8%, and 62.7% after 1, 2, 5, and 10 years, respectively.

## 4. Discussion

The medical records of a large number of SLE patients with long-term follow-up were collected in this retrospective cohort study. This study is a pivotal study investigating the impact of the presence of lupus pericarditis on mortality even in the absence of cardiac tamponade. Moreover, SLE patients exhibited a high risk of pericarditis, which was consistent with the finding of a previous study [7]. SLE patients with pericarditis are likely to develop cardiac tamponade, which can be fatal. Furthermore, SLE accompanied by pericarditis is a severe inflammatory condition that requires high-dose steroid therapy, which can result in severe secondary infections and mortality. 

According to Badui et al., pericarditis is the most common cardiovascular complication, which was found in 39% of 100 consecutive female patients with active SLE [7]. Moreover, the study similarly found a high proportion of SLE patients with pericarditis (16%). However, previous research on lupus pericarditis is limited, and the majority of findings were reported before 2010. Nevertheless, some studies have suggested that the presence of large pericardial effusion and impending cardiac tamponade may result in a poor prognosis. Goswami et al. reported that lupus pericarditis was detected in 25.4% of 409 SLE patients in India and mentioned that pleuritis, anti-nucleosome antibody, and the size of pericardial effusion (used to predict the development of tamponade) were all factors that increased the risk of cardiac tamponade [11]. Consistent with previous study findings, our study findings revealed nearly the same proportion of patients with lupus pericarditis (16.4%). In addition, our study indicated that lupus pericarditis could still affect survival even when cardiac tamponade was not present. This study also showed that a large proportion of SLE patients with lupus pericarditis had high disease activities, low C3 levels, and multiple comorbidities. Moreover, after adjustment for age, sex, disease activity, and comorbidities in the multivariate analysis, lupus pericarditis was found to be significantly correlated with the increased risk of mortality. However, several studies have described pericarditis as a benign sign. Estes et al. in 1971 reported that 20% of SLE patients had pericarditis, with only two patients developing cardiac tamponade and no deaths reported [12]. Kasitanon et al. also once mentioned that central nervous system (CNS) lupus, lupus nephritis, pneumonitis, and myocarditis were the primary causes of SLE-related death [13]. Nevertheless, as the clinical diagnosis of lupus myocarditis is challenging, patients with lupus pericarditis cannot be verified to have myocarditis based only on their medical records. Despite this, lupus pericarditis can still be considered as a highly inflammatory condition that could lead to fulminant heart failure and death [14]. According to the SLE cardiac pathology review by Jain et al., pericardial involvement was found in 43–83% of autopsy cases, with approximately 25% exhibiting clinical symptoms [15].

Fibrinous pericarditis is the most prevalent type of lupus pericarditis. Long-term pericarditis can cause adhesions between the visceral and parietal layers, especially obliterating the pericardial space, which can either be focal or diffuse [16]. Moreover, direct immunofluorescence studies on lupus pericarditis tissue showed a granular deposition of immunoglobulin, C1q, and C3 in the walls of pericardial vessels. Inflammation and immune reactants were detected in close proximity to the examined tissues [17]. These studies have reported that the pathogenesis includes severe inflammatory changes and immune complex deposition. The pathogenesis of lupus pericarditis is thought to be responsible for systemic organ inflammation and mortality. Several previous studies have also examined the factors that may increase mortality risk in SLE patients. Further studies are still needed to explain the pathogenesis of lupus pericarditis and how it may increase the risk of mortality.

Several studies also evaluated the risk factors for mortality in SLE patients. Lupus nephritis and CNS lupus are conventional risk factors. According to Ward et al., nephritis (relative risk, 2.34) and seizures (relative risk, 1.77) were associated with poor overall survival [18]. Moreover, a study found that SLE patients had a significantly higher mortality rate resulting from vascular disease, especially accelerated atherosclerosis. Additionally, both disease and therapeutic modalities, particularly treatment with corticosteroids, appear to contribute to the high prevalence of coronary artery disease [19]. However, Fatemi et al. found that both pericarditis and seizure present at the time of SLE diagnosis significantly reduced the survival rate. Additionally, no reported case of pericarditis showed a decrease in the risk of mortality (HR = 0.22) [20]. Even though Mittoo et al. observed a nearly 2-fold higher prevalence of pleuritis (OR = 1.98, 95% CI = 1.31, 2.82), pericarditis was not included in their paper [21]. Pamuk et al. found that serositis at the time of diagnosis, SLEDAI score 6, and autoimmune hemolytic anemia were independent prognostic factors for survival in 428 SLE patients in Turkey [22]. By contrast, our study used the retrospective medical records to identify SLE patients who presented with lupus pericarditis after admission. Several prior investigations have shown similar findings, all indicating that lupus pericarditis would increase the risk of mortality.

This study has several limitations. First, the retrospective nature of this study may lead to some limitations, such as incomplete data and inconsistencies in the temporal relationship between dependent and independent variables. Second, the causes of pericarditis may be misclassified because lupus pericarditis was diagnosed only based on its clinical manifestations. Third, information on some possible confounders, such as socioeconomic levels and smoking behaviors, was not available. Despite these limitations, the strength of this study is that it is a large-scale cohort study with long-term follow-up.

## 5. Conclusions

Lupus pericarditis is a common occurrence in SLE patients. Cardiac involvement is one of the most common complications, resulting in elevated risks of morbidity and mortality. Patients with lupus pericarditis should be treated with caution even if there is no concomitant cardiac tamponade, although pericarditis is not regarded as primary organ involvement in lupus. The study also revealed that patients with SLE have high risks of pericarditis and mortality.

## Figures and Tables

**Figure 1 jcm-11-05473-f001:**
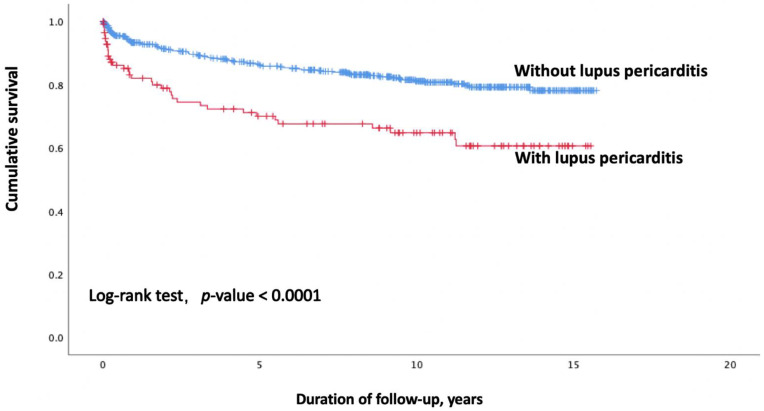
Survival curves of SLE patients with and without lupus pericarditis. Abbreviations: SLE, systemic lupus erythematosus.

**Table 1 jcm-11-05473-t001:** Baseline characteristics of SLE patients with and without lupus pericarditis.

	All Patients (*n* = 689)		With Pericarditis (*n* = 113)		Without Pericarditis (*n* = 576)		*p*-Value
Age at admission, mean (SD)	40.78	15.95	41.57	18.53	40.62	15.40	0.0165 *
Female, number (%)	611	88.70	99	87.61	512	83.80	0.695
Follow-up duration by year, mean (SD)	7.31	5.11	6.44	5.48	7.55	5.02	0.048 *
Disease activity and serology, mean (SD)							
SLEDAI score	12.92	6.34	16.75	7.25	12.17	5.87	0.0001 *
WBC count (μL)	6304.85	3891.67	6683.19	3798.59	6230.63	3908.61	0.251
Hemoglobin (mg/dL)	11.30	5.30	10.17	2.36	11.53	5.68	0.013 *
PLT (k/μL)	20.25	16.23	20.21	10.57	20.26	17.12	0.977
Anti-dsDNA titer (WHO unit/mL)	441.61	598.80	519.49	691.00	426.37	578.59	0.132
C3 (mg/dL)	63.38	32.85	60.85	32.28	69.84	32.78	0.008 *
C4 (mg/dL)	13.59	9.59	12.38	8.39	13.83	9.79	0.142
Comorbidity, number (%)							
Diabetes	50	7.3	6	5.3	44	7.7	0.383
Hypertension	220	31.9	44	40.0	176	30.6	0.201
Cardiovascular event	55	8.0	19	16.8	36	6.3	0.0001 *
End-stage renal disease	66	9.6	19	16.8	47	8.2	0.004 *
Medication use, number (%)							
Steroid	677	98.3	111	98.2	566	98.2	0.98
Hydroxychloroquine	688	99.9	83	73.5	442	76.7	0.435
Mycophenolic acid	78	11.3	20	17.7	58	10.1	0.019
Cyclophosphamide	102	14.8	25	22.1	77	13.4	0.017
Azathioprine	268	38.9	52	46.0	216	37.5	0.092

SLE, systemic lupus erythematosus; SLEDAI, systemic lupus erythematosus disease activity index; WBC, white blood cell; Anti-dsDNA, anti-double strand DNA; C3, complement 3; C4, complement 4; SD, standard deviation; %, percentage; μL, microliter; dL, deciliter; mg, milligram; k, kilo; * *p* < 0.05.

**Table 2 jcm-11-05473-t002:** Multivariate Cox regression analysis: adjusted hazard ratio of the risk factors for mortality in SLE patients.

Risk Factor	Hazard Ratio	95% Confidence Interval	*p*-Value
Male vs. female	1.294	0.771–2.173	0.329
Age	1.053	1.040–1.065	<0.0001 *
Lupus pericarditis	1.963	1.315–2.963	0.001 *
Cardiovascular event	0.572	0.313–1.045	0.069
End-stage renal disease	2.533	1.620–3.961	<0.0001 *
SLEDAI score	1.079	1.043–1.116	<0.0001 *
Anti-dsDNA titer	1.000	1.000–1.000	0.523
C3	1.000	0.993–1.116	0.995
C4	1.004	0.987–1.023	0.8353

SLEDAI, systemic lupus erythematosus disease activity index; Anti-dsDNA, anti-double strand DNA; C3, complement 3; C4, complement 4; SD, standard deviation; * *p* < 0.05.

## Data Availability

Data cannot be shared for ethical/privacy reasons.

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
