# Peer review of "Exploring the Risk Factors for Poor Survival in Lupus Pericarditis Patients: A Retrospective Cohort Study"

_jcm, 2022, doi:10.3390/jcm11185473_

Round 1
Reviewer 1 Report
I really enjoyed reading this journal.
However, there are a number of points I have raised and included in the attached pdf as comments.

Author Response
Editor in Chief, Journal of Clinical Medicine
Dear Editor and Reviewers:
Thank you for reviewing our manuscript (Manuscript ID JCM-1901192 “Exploring the risk factors for poor survival in lupus pericarditis patients: A retrospective cohort study”). The reviewer comments are greatly appreciated. We have major revised the manuscript following reviewers’ comments. We replied reviewers’ comments point-to-point as below. All the revised text was in red for easy reading.
To Reviewer 1:
Reviewer: 1
1. Are all these lupus patients? Please clarify in the text
Answer: we have revised in line 27 ‘’Cox regression adjusted analysis indicated that lupus pericarditis (hazard ratio = 1.963, 95% CI = 1.315, 2.963, p = 0.001)…’’
- Compared to what group? Those without pericarditis?
Answer: we have revised in line 31 to 33 ‘’Moreover, the Kaplan–Meier survival curve analysis revealed that patients with pericarditis compared without pericarditis had a higher mortality rate (log-rank test, p < 0.001)’’.
- Please add a reference here.
Answer: Thanks for reminding. We added reference at line 57 “In addition, previous studies have mostly found that patients with cardiac tamponade have a higher mortality rate [6,9].’’
- Not compulsory - but it would be nice to see the reasons why patients with lupus were admitted (presenting complaints) in a table
Answers: Most lupus patients was admitted due to infection or disease progression. But most important of all is more detail medical records after admission.
- Please update table to add parentheses where they are needed around SD and % to make it easier to follow.
Answer: We have added parentheses on Table 1 at line 121 ‘’SLE, systemic lupus erythematosus; SLEDAI, systemic lupus erythematosus disease activity index; WBC, white blood cell; Anti-DsDNA, Anti-double strand DNA; C3, Complement 3; C4, Complement 4; SD, standard deviation; ; %, percentage; μL, microliter; dL, deciliter; mg, milligram; k, kilo; *, p<0.05.’’
- Table 2?
Answer: Thanks for reminding. We revised it to Table 2.
- Moreover used twice in succession. Does not read well.
Answer: We have deleted moreover at line 166.
- ?
Answer: We have added at line 189 to 190 ‘’However, several studies have described pericarditis as a benign sign.’’.

Reviewer 2 Report
Lupus pericarditis is the most common type of heart disease affecting those with lupus and definitely related research needs more attention. However, in this manuscript, many questions are left hanging.
1. There is a significant portion of patients who have diabetes. Both SLE and diabetes could lead to pericarditis and ESRD respectively and a deep dive to analyze these two groups separately will help improve this paper.
2. Sudden death in lupus patients is uncommon and it would be interesting to know how many cases of sudden death due to myocardial infarction have been recorded.
3. The cardiac involvement as the initial manifestation of SLE is rarely reported. It would be interesting to know how many patients had lupus pericarditis at admission and how many patients developed pericarditis during the follow-up. Do these two groups display different disease characteristics and have different survival rate?
Author Response
To Reviewer 2:
Reviewer: 2
Lupus pericarditis is the most common type of heart disease affecting those with lupus and definitely related research needs more attention. However, in this manuscript, many questions are left hanging.
- There is a significant portion of patients who have diabetes. Both SLE and diabetes could lead to pericarditis and ESRD respectively and a deep dive to analyze these two groups separately will help improve this paper.
Answer: Thanks for your valuable comments. In deed, both SLE and diabetes can lead to pericarditis. In our analysis, we have make sure the all patients with pericarditis are contributed by lupus. The description can be found at line 84 to 87 “and diagnostic signs of enlargement of the cardiac silhouette on a chest radiograph together with the echocardiographic diagnosis of pericardial effusion, as well as exclusion of other causes of pericarditis, such as tuberculosis or malignancy.”
- Sudden death in lupus patients isuncommon and it would be interesting to know how many cases of sudden death due to myocardial infarction have been recorded.
Answer: In our study, 4 of the 113 patients were represented with cardiac tamponade as the presentation of systemic lupus erythematosus during observation period. The causes of mortality were infection in three cases and only one case died from cardiac tamponade induced sudden death. To clarify the outcomes, we have revised to line 133 to 136.
- The cardiac involvement as the initial manifestation of SLE is rarely reported. It would be interesting to know how many patients had lupus pericarditis at admission and how many patients developed pericarditis during the follow-up. Do these two groups display different disease characteristics and have different survival rate?
Answer: After reviewing our medical records, there are 102 of 689 patients (14.8%) with initial presentation of lupus pericarditis. The rest of 11 patients with lupus pericarditis were developed during follow-up period. I have done some simple K-M survivial curve to compare the mortality difference. Red line is initial presentation with lupus pericarditis and green line is lupus pericarditis during follow-up. Initial lupus pericarditis group compared with lupus pericarditis during follow-up showed no survival difference (Figure 1). Therefore, we think we need to focus on the presentation of lupus pericarditis. Whether lupus pericarditis started at beginning or successive during follow-up. Lupus pericarditis is a poor prognosis factor.
Figure 1. Survival curves of SLE patients without lupus pericarditis, patients with initial presentation of lupus pericarditis and patients with lupus pericarditis during follow-up. Abbreviations: SLE, systemic lupus erythematosus

Round 2
Reviewer 2 Report
All my questions have been addressed.